# Strengthening Surveillance Systems for Malaria Elimination by Integrating Molecular and Genomic Data

**DOI:** 10.3390/tropicalmed4040139

**Published:** 2019-12-03

**Authors:** Christian Nsanzabana

**Affiliations:** 1Department of Medicine, Swiss Tropical and Public Health Institute, 4051 Basel, Switzerland; christian.nsanzabana@swisstph.ch; Tel.: +41-61-284-82-52; 2University of Basel, P.O. Box, CH-4003 Basel, Switzerland

**Keywords:** malaria, surveillance, molecular, genomic, *Plasmodium falciparum*, *Plasmodium vivax*, sequencing

## Abstract

Unprecedented efforts in malaria control over the last 15 years have led to a substantial decrease in both morbidity and mortality in most endemic settings. However, these progresses have stalled over recent years, and resurgence may cause dramatic impact on both morbidity and mortality. Nevertheless, elimination efforts are currently going on with the objective of reducing malaria morbidity and mortality by 90% and malaria elimination in at least 35 countries by 2030. Strengthening surveillance systems is of paramount importance to reach those targets, and the integration of molecular and genomic techniques into routine surveillance could substantially improve the quality and robustness of data. Techniques such as polymerase chain reaction (PCR) and quantitative PCR (qPCR) are increasingly available in malaria endemic countries, whereas others such as sequencing are already available in a few laboratories. However, sequencing, especially next-generation sequencing (NGS), requires sophisticated infrastructure with adequate computing power and highly trained personnel for data analysis that require substantial investment. Different techniques will be required for different applications, and cost-effective planning must ensure the appropriate use of available resources. The development of national and sub-regional reference laboratories could help in minimizing the resources required in terms of equipment and trained staff. Concerted efforts from different stakeholders at national, sub-regional, and global level are needed to develop the required framework to establish and maintain these reference laboratories.

## 1. Background

Malaria remains a major public health problem globally. In 2017, there were approximately 219 million cases and 435,000 related deaths worldwide, with most cases and deaths in young children in sub-Saharan Africa [1]. Nevertheless, there was a tremendous decrease in malaria morbidity and mortality between 2000 and 2015, the number of cases and related deaths decreasing by 18% and 48%, respectively [2]. At the same time, the number of countries with ongoing malaria transmission decreased from 106 to 95, while the number of countries progressing towards malaria elimination, i.e., <1000 cases/year, increased from 13 to 33 [2]. All those gains were possible due to unprecedented efforts in malaria control by the provision of efficacious control measures including long-lasting insecticide-treated bed nests (LLINs) and artemisinin-based combination therapies (ACTs). Consequently, the hope that a global eradication is feasible in the foreseen future with sustained investment in malaria control remerged, and a strategy of decreasing malaria prevalence by 90% globally and elimination of malaria in at least 33 countries was adopted in 2015 by the World Health Organization (WHO) [3]. However, this progress has stalled over the last few years, with morbidity and mortality increasing again in some countries between 2015 and 2017 [1]. Therefore, to reach those ambitious goals, surveillance systems need to be strengthened, and new tools are needed to improve diagnosis of low parasite densities, improve the characterization of the parasite reservoir, and better assess transmission to provide precise and robust data for surveillance [4,5]. Indeed, currently malaria diagnosis is mainly based on light microscopy (LM) and rapid diagnostic tests (RDTs); however, those tools, even though very useful, have some limitations in the elimination context. Recently, highly sensitive RDTs (HS-RDTs) have been developed and have been shown to improve performance in detecting low-density infections compared to LM or RDTs [6,7,8]. Nevertheless, more sensitive diagnostic tools may be required to detect transmissible very low parasites densities [9,10,11], and the emergence and spread of parasite with HRP2/3 deletion could definitely compromise the use of this new HS-RDT in some settings [12,13].

By increasing sensitivity more than 10,000× compared to LM and RDTs, molecular techniques provide a more sensitive approach to diagnose very low parasite density infections [14], allowing to detect asymptomatic carriers who may contribute largely to transmission [15,16,17]. However, in elimination settings, detecting very low-density parasites is not enough to assess transmission at low-level scale and the impact of interventions. Genomic data may provide a more detailed picture on the parasite population structure and fine-scale data that could help customizing interventions [18]. Indeed, genomic data can provide information about the parasite population structure at local level, allowing for example to differentiate local from imported parasites, detecting foci of transmission, characterizing connectivity and transmission of different parasite strains [19,20]. Whole-genome sequencing (WGS), targeted amplicon deep sequencing (TADS), and other single nucleotide polymorphisms (SNPs)-based genotyping assays such as high-resolution melting (HRM) are very powerful tools to assess those dynamics. Indeed, sequencing costs are decreasing, and novel techniques have been established to improve and standardize sequencing quality, even from dried blood spots (DBS), facilitating the use of the technique in epidemiological and surveillance studies [21,22,23]. Moreover, new analytical methods have also been developed to facilitate the analysis and provide meaningful results [24,25,26]. The integration of epidemiological, molecular, and genomic data could substantially improve malaria surveillance in elimination settings [27,28,29]. However, until now, those tools have been essentially used in research settings and rarely in routine surveillance. In this review, we describe the advantages and disadvantages of those new techniques and provide suggestions on how they could be integrated in routine surveillance in pre-elimination or elimination settings. 

## 2. Molecular Diagnosis

Various molecular techniques are currently used for malaria diagnosis: Nested polymerase chain reaction (nPCR), quantitative PCR (qPCR), quantitative reverse transcriptase PCR (qRT-PCR), loop-mediated isothermal amplification (LAMP), and quantitative nucleic acid sequence-based amplification (QT-NASBA) [30]. The different techniques with their specific characteristics have their advantages and disadvantages and may be appropriate for specific applications (Table 1). However, there is clearly a lack of highly sensitive/high throughput point-of-care (POC) molecular test [5], even though LAMP may be considered currently as the best available alternative to be used as a molecular diagnostic POC tool [31]. Different LAMP technologies have been developed, and sensitivity can go as low as 1 to 10 parasites/µL [32]. PCR-based techniques are more sensitive, increasing the sensitivity up to 0.022–0.1 parasites/µL [33,34], 0.016 parasites/µL by RT-qPCR [35], or even 0.003362 parasites/µL by amplifying total nucleic acids (DNA and RNA) with RT-qPCR [14]. However, those techniques are better suited for well-equipped laboratories, with well-trained staff [36]. Those techniques may be better indicated for large cross-sectional surveys, for example before and after a campaign of mass drug administration (MDA) for baseline surveys and impact assessment [37,38,39]. New techniques are still in development and could complete the portfolio of molecular methods available for malaria diagnosis, for example, the PCR nucleic acid lateral flow immunoassay (NALFIA) [40] and microfluidics-based assays [41], and could potentially improve the sensitivity and throughput, while providing POC molecular diagnostic testing alternatives.

## 3. Population Genetics

Populations’ genetic studies for human malaria parasites (*P. falciparum* and *P. vivax*) to assess the dynamics of transmission or the relatedness of parasite populations have for long time used microsatellites [28,44,45,46]. Subsequently, other techniques, such as molecular barcode based on genotyping of a limited number of SNPs by HRM [29,47,48], high throughput SNPs genotyping on the Sequenom MassAarray iPLEX platform, or large scale SNPs genotyping using a custom 384-SNP Illumina GoldenGate platform [49,50], have been used. However, the optimal use of microsatellites is limited to monoclonal infections, and PCR amplifications of those short tandem repeats often produce artefacts that make the data analysis complicated, while the SNP barcode by HRM is limited in detecting minority clones [36,51]. Recent studies have used either WGS [27] or TADS [52] to assess those dynamics on a finer scale. WGS offers the advantage of providing a full picture of the parasite genome and is more suitable for studying foci of transmission with reduced diversity and mainly clonal parasites, as the detection of minority clones is limited [53]. WGS is also important in detecting new genetic variants associated with antimalarial drug resistance, when molecular markers associated with antimalarial drug resistance have not yet been discovered or validated [54,55], and may be useful to provide a population prevalence of G6PD deficiency in regions where radical cure of *P. vivax* with 8-aminoquinolines is required [56]. The use of TADS has the advantage of reduced cost and work by sequencing a defined genome target, thus allowing the use of very low starting DNA material, such as those usually collected from routine surveys on dried blood spots [57] or even from RDTs [58]. TADS also has the advantage of detecting minority clones and reconstructing haplotypes in multiple clones infections [52,59]. The different techniques may be used for different applications in pre-elimination and elimination settings (Figure 1). 

## 4. Discussion

Malaria elimination and eradication are back on the agenda of the global health community, and discussions about their feasibility are going on. The WHO in its global technical strategy 2016–2030 is aiming at eliminating malaria in 30 countries by 2030 [3]. To reach those ambitious targets, there is a need to reassess the current strategies for malaria control and optimize them. One of the main changes to move from control to elimination is to include surveillance as a core component in any malaria control program [60]. Recent success in malaria elimination, such as in Sri Lanka, have shown that well-trained staff, political commitment, and a good surveillance system are key for a successful malaria elimination program [61]. 

Case detections in malaria elimination settings are still mainly based on LM and RDT. Even though those techniques have proved to be successful in some malaria elimination settings, the addition of molecular techniques could greatly improve the sensitivity of parasites detection, especially for asymptomatic cases [62]. qRT-PCR is currently the most sensitive technique for malaria diagnosis but is not practical due to the inherent difficulty in working with RNA, even in well-equipped laboratories [35]. Therefore qPCR, especially highly sensitive qPCR [9,33,34], would be more suitable for malaria diagnosis in reference laboratories with good infrastructure and well-trained staff (Figure 2). Those laboratories could be at national level for small countries but could be also at sub-national level in bigger countries. Ultrasensitive qPCR techniques using finger prick blood volume for *P. falciparum* [33] and *P. vivax* [63] would be easier to implement compared to the high volume qPCR that is more sensitive but would be less practical in routine surveillance due to the venipuncture blood [9,34]. Ultrasensitive qPCR could be used, for instance, for baseline surveys before MDA or to assess the impact of different interventions [16,38]. QT-NASBA and RT-qPCR would have the advantage of being able to detect gametocytes, for example in studies evaluating the efficacy of gametocytocidal drugs, or to evaluate the impact of interventions aiming at reducing transmission [64,65,66]. Those different interventions do not require prompt results for case management; therefore, the analysis could be easily conducted in a sub-national or national reference laboratory. This will ensure good quality data as well, as the laboratories will have time to perform the experiments with good quality standards. However, point-of-care tests are lacking for strategies such focal screening and treatment (FSAT) or mass screening and treatment (MSAT). In the instance LAMP could be used, even though, the low throughput and the lower sensitivity of the technique compared to other molecular techniques would probably not provide optimal results, especially for asymptomatic cases. LAMP techniques could be used as well at point of care and could help detect asymptomatic carriers in passive case detection (PCD), active case detection (ACD), and reactive case detection (RACD). As LAMP techniques are more sensitive than microscopy and RDTs, they could be in some cases a useful addition to the diagnostic tool box for very low endemic settings [31,32]. However, new techniques in development, such as microfluidics-based nucleic acid tests, would be better suited for population-based interventions such as FSAT and MSAT, as they would have the advantage of increasing the sensitivity, improving the throughput, and in some instances if they are multiplexed, could provide information on non-malaria infection for prompt patient care management [41]. 

One of the recommendations of the malERA expert panel is to improve the characterization of the parasite reservoir and develop new tools to assess transmission [4]. Those tools are available and are being progressively deployed in malaria endemic countries [67]. Currently HRM genotyping using SNPs offers a relatively low-cost, robust, and easy to implement method to study parasite population structure and molecular markers associated with antimalarial drug resistance [68,69]. The technique could easily be implemented in laboratories in malaria endemic settings, even though it would need to be implemented only in reference laboratories with qualified personnel and adequate infrastructure. However, NGS methods could potentially provide more information, especially to study parasite population structure and dynamics at lower scale level [25,70]. WGS does provide complete information about the parasite genetics, but it may not be needed in most cases. WGS would be more appropriate in pre-elimination settings to provide insights into the local and/or regional parasite populations to guide the selection of the right markers to be used in the elimination phase, and TADS would provide the most useful information in elimination settings and would be more cost-effective (Figure 1). The development of long-read sequencing platforms, such as Oxford Nanopore Technology (ONT) and Pacific Biosciences (PacBio), could help improve the usefulness of sequencing data for surveillance. Though currently limited by their higher error rate compared to Illumina sequencing, their capacity of providing long reads up to 10 kb [71,72,73] could be useful in developing long amplicon reads combining molecular markers of resistance and genetic diversity SNPs. Moreover, the combination of long- and short-reads sequencing data in the hybrid assembly has been show to improve substantially the accuracy of the data [74]. The introduction of genomic data could also improve surveillance by providing more sensitive and accurate data on transmission, vector and drug resistance, differentiating imported from local infections, or determining connectivity of parasite populations [18]. Integrated with epidemiological data, the genetic information could help in detecting foci of transmission and customizing the interventions. Indeed, this will give information on the dynamics of parasite population in a specific region, allowing better targeted interventions. For example, in countries with *P. vivax*, the radical cure of hypnozoites is of great importance. Effective drugs such as primaquine and tafenoquine are available; however, their use is limited by the lack of point-of-care test for G6PD-deficient patients [75]. Population-based surveys looking at the different mutation variants associated with G6PD deficiency could provide useful baseline data for radical cure treatment policy in the absence of a validated POC test [76]. Combined with modelling, the genomic information could help predict the distribution and dynamics of the parasite population and the spread of resistant parasites and mosquitoes, allowing for informed and tailored interventions. To get reliable and complete information from genomic data, sample sizes must be adequate for the specific questions asked; the centralization of samples storage collected during routine surveillance or case management at reference laboratories could help in having access to a large number of samples to analyze if the appropriate metadata are also stored appropriately (Figure 2). Data transfer systems using mobile phone-based applications could be implemented at the different levels of the health systems for accurate and rapid transmission of information to the reference laboratories and the National Malaria Control Program [77,78,79]. 

## 5. Outlook

Globally, a consensus has been established that good quality data are required to improve disease surveillance, especially in low- and middle-income countries (LMICs) [80]. The establishment of well-equipped reference laboratories with well-trained staff in most malaria endemic settings will require concerted efforts from different stakeholders, and several challenges ranging from funding to staff retention and procurement of required reagents are only a part of them [67,81,82]. However, the use of NGS could be centralized to very few sub-regional reference laboratories located in malaria endemic countries [36,83,84]. This requires a lot of effort to develop the capacity not only for sequencing but more importantly for data analysis, a critical part in the path for developing countries to get access to these techniques [85,86,87]. This should be a concerted effort not only from the malaria community but from the global health community, as other diseases such as tuberculosis (TB), human immunodeficiency virus (HIV), and other bacterial and viral infections may benefit as well from these centers of excellence [88]. The private sector should be involved to support the development of data sharing and storage platforms, and that will require clear regulations to be set up by local authorities on data privacy and ownership [89,90]. Support from well-established and expert laboratories in developed countries would be required, and establishment of sustainable external quality assurance (EQA) programs will be of paramount importance [91]. The development of sub-regional center of excellence will also require the establishment of samples and data sharing policies between different countries, and initiatives such as the Malaria Genomic Epidemiology Network (MalariaGEN) and Plasmodium Diversity Network Africa (PDNA) are providing examples of successful platforms for collaboration and data sharing [92,93]. It is also crucial that national malaria control programs (NMCPs) are playing a central role in these changes (Figure 2), and they must develop clear policies based on their needs and resources, provide clear guidance to the collaborating laboratories and the different key stakeholders at each level of the health system, and coordinate the different activities. With decreasing transmission in some settings and not neighboring regions, imported malaria cases will become an increasing threat to elimination, forcing different countries to cooperate. In China for example, almost all malaria cases are now imported, not only from neighboring countries, but also from Africa [94]. Therefore, it should be easier to convince the malaria stakeholders that approaches at the different levels are required, and sub-regional centers of excellence for genomics analyses could be a cost-effective way of improving collaborations between countries for the same goal of malaria elimination and eradication.

## Figures and Tables

**Figure 1 tropicalmed-04-00139-f001:**
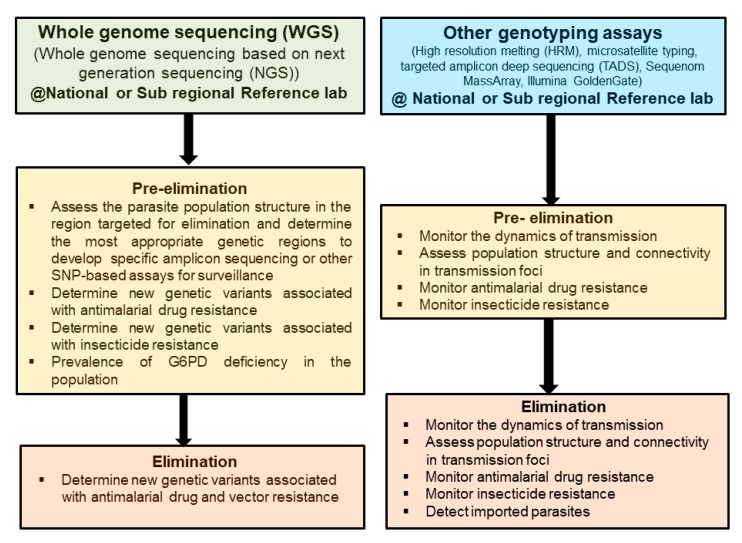
Applications for the different genomic techniques for malaria surveillance in pre-elimination and elimination settings. A cost-effective way of using whole-genome sequencing would be to use it mainly in pre-elimination settings to assess the population genetic structure to help develop customized gene-targeted assays to use in elimination settings.

**Figure 2 tropicalmed-04-00139-f002:**
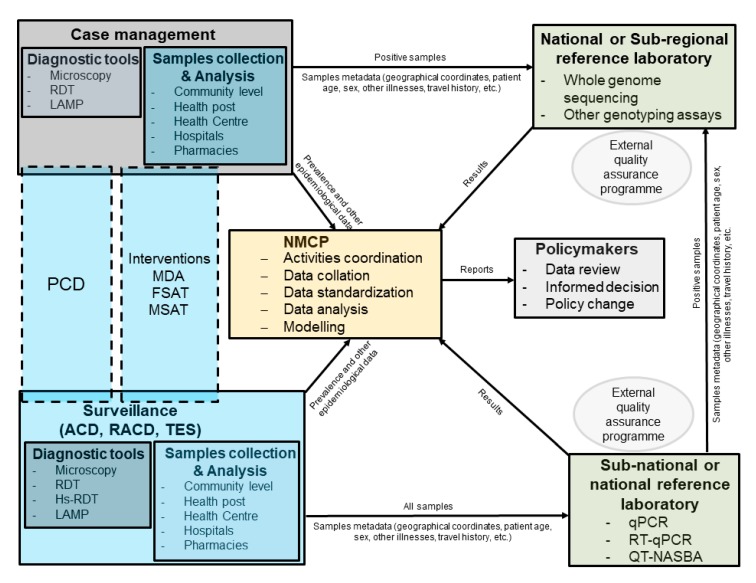
Schematic representation describing how reference laboratories for molecular diagnostic and genomic analyses could be integrated into routine surveillance for malaria elimination. The national malaria control program (NMCP) should coordinate the different activities. All positive samples collected during routine case management at health facility or by community health workers (CHWs) should be sent directly to the reference laboratory. Samples collected during surveillance should be send to regional or national reference laboratory, where they are screened for malaria, and positive samples would be then sent to the national or sub-regional reference laboratory. Metadata collected during case management of surveillance should be sent directly to NMCP in real-time by apps developed specifically for malaria surveillance.

**Table 1 tropicalmed-04-00139-t001:** Characteristics of the different molecular diagnostic techniques used to detect malaria parasites.

Assay	Limit of Detection (Parasites/µL)	Throughput	Cost/Sample Excluding Labor and Equipment (USD)	Advantages	Disadvantages	Reference
**Nested PCR**	1	moderate	<10	Requires simple and cheap thermocycler Can be performed with low amount of DNA (e.g., from dried blood spots)	Moderately sensitive Requires good laboratory infrastructure and well-trained staff	[42]
**qPCR** **(high blood volume)**	0.022	high	<10	Highly sensitive	Requires high blood volume Requires good laboratory infrastructure and well-trained staff	[34]
**qPCR** **(low blood volume or dried blood spots)**	0.15	high	<10	Highly sensitive Can be performed with low amount of DNA (e.g., from dried blood spots)	Requires good laboratory infrastructure and well-trained staff	[33]
**qRT-PCR**	0.002	high	<20	Highly sensitive Can be performed with low amount of DNA (e.g., from dried blood spots) Can detect and quantify gametocytes	Difficult to work with RNA Requires good laboratory infrastructure and well-trained staff	[14]
**LAMP**	1 to 5	moderate	<3	Cheap Does not require laboratory infrastructure and well-trained staff Can be performed with low of DNA (e.g., from dried blood spots) or directly from blood sample Fast	Moderately sensitive Limited throughput	[31]
**QT NASBA**	<1	high		Can be performed with low amount of DNA (e.g., from dried blood spots) Can detect and quantify gametocytes	Not as robust as qRT-PCR	[43]

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
