# Peer review of "Strengthening Surveillance Systems for Malaria Elimination by Integrating Molecular and Genomic Data"

_tropicalmed, 2019, doi:10.3390/tropicalmed4040139_

Round 1
Reviewer 1 Report
The manuscript entitled " Integration of molecular and genomic data in routine surveillance for malaria elimination" is a review of molecular and genomic techniques. The review is useful for one that wants to know the molecular and genomic techniques used in the surveillance of malaria.
There are few points to change/correct
1 - Abstract - page 1 - line 15
Please replace "Polymerase"
2- Table 1 - page 92
There is a black line on the top of the table.
3 - Figure 1
Please improve figure 1, some words are distorted.
4 - I think that outlook (perspectives) could be a separate section.
Reviewer 2 Report
Honestly speaking, I read the manuscript with much interest. But in the end I found myself asking some questions. What is (are) the main motivating question(s) of the authors behind this manuscript? And, how did they have approached to answer their motivating question(s)? I struggled to get my answer(s) straightforwardly. In other words, I am saying that the manuscript needs revising before I could be able to make recommendation for its publication.
My comments are as follows:
1) I strongly suggest the authors to restructure some opening paragraphs of their manuscript in order for helping potential readers, like myself, to find the motivating question(s) and their approaches towards answering the motivating question(s). I must say, the authors have them. For example, the authors say, in line 6-7, "... these progresses have stalled over the last 4 years, and resurgence may cause dramatic impact on both morbidity and mortality." Although the authors need to cite a credible source to back up their statement that [these progresses have stalled over the last 4 years], these statements in Abstract do not appear in Background. The repeat of these statements in Background could have made easy for the authors to argue for their suggestions that appear in lines 70-72.
2) I also feel that the title is not suitably crafted to reflect what the manuscript stands for. According to my reading of the manuscript, the authors are arguing for strengthening malaria surveillance systems, regionally or at country levels, in pre-elimination and/or elimination settings. The title simply appears to be dull in conveying that message clearly.
3) The authors cite a case study of Sri Lanka. They say, "Recent success in malaria elimination, such as Sri Lanka have shown that well-trained staff, political commitment and a good surveillance system are key for a successful malaria elimination programme [61]." In the conclusion, the authors do not seem to argue for the importance of political commitment, which, I think, is an important element for achieving malaria elimination from a country or from a region, and, therefore, must be part of and be incorporated in any discussion about malaria elimination programmes. In addition, could the authors say with some confidence that Sri Lankan surveillance system did have the new tools (i.e., molecular and population genetics diagnostic tools - the authors appear to be strong advocate of in this manuscript)?
4) The authors have not described (or, more appropriately, not walked us through) the contents of Figure 1 or 2. The figure captions are very brief and thus does not help much.
5) Table 1 is good, but the row-wise contents appear to be misplaced in different columns. For some reason, I could not see/read what are written in the column names (in the first row) because of its pitch dark color. For qPCR (High blood volume), the 6th column contains "- Requires good laboratory infrastructure and well-trained staff". But for the RT-qPCR row, the 5th column has "- Does not require laboratory infrastructure and well-trained staff".
6) The language of the manuscript appear to be good, with a few exception. Why it is per example? Why not say, for example? Also, in line 139 has should be replaced with "have" while in line 148 it should be "... point of care test is ..."
Reviewer 3 Report
- Other review articles have covered the idea of utilizing NGS technology and insights for surveillance methods and elimination strategies (https://malariajournal.biomedcentral.com/articles/10.1186/s12936-019-2880-1, https://link.springer.com/article/10.1007%2Fs00436-018-6127-9), but the submitted manuscript has more focus on the integration of several different techniques and how to best utilize and integrate all of the data, which is useful information to consider.
- This appears to have been submitted as an article, but it should be submitted as a review article. The abstract refers to the manuscript as a review, so it may have just been submitted wrong. Some of the ideas of how to integrate data networks (Fig 2) are new to this study, but is really just further developing an idea by considering existing literature, and is not a “scientifically sound experiment that provides a substantial amount of new information”, as required for article submissions to this journal.
- The headings for table 1 are not visible to the reader, and rows 5 and 6 could be stretched (at the expense of other rows) so that the table is not as tall.
Formatting issues:
- Abstract is too long by 100 words.
- Some minor spelling and grammar issues throughout
- Some paragraphs are very long and should be split at logical positions.
Round 2
Reviewer 2 Report
I have read the authors' responses to my comments on the manuscript version that I reviewed. The revised manuscript has improved significantly as a result of the authors' inclusion of many of the responses. I am happy to make a recommendation of its publication.
Reviewer 3 Report
The authors have addressed the reviewer concerns